# Beneficial Effect of Dietary Approaches to Stop Hypertension Diet Combined with Regular Physical Activity on Fat Mass and Anthropometric and Metabolic Parameters in People with Overweight and Obesity

**DOI:** 10.3390/nu16183187

**Published:** 2024-09-20

**Authors:** Małgorzata Soroń-Lisik, Paweł Więch, Mariusz Dąbrowski

**Affiliations:** 1Institute of Health Sciences, College of Medical Sciences, University of Rzeszów, 35-310 Rzeszów, Poland; malgorzataso@dokt.ur.edu.pl (M.S.-L.); pwiech@ur.edu.pl (P.W.); 2Institute of Medical Sciences, College of Medical Sciences, University of Rzeszów, 35-310 Rzeszów, Poland

**Keywords:** overweight, obesity, DASH diet, physical activity, anthropometric parameters, metabolic parameters

## Abstract

Background/objective: Obesity and overweight have become growing health-related issues worldwide, which also applies to Poland. Excess fat mass is associated with an increased risk of metabolic and non-metabolic complications. The aim of our pre–post-designed study was to assess the effect of behavioral intervention on body weight, fat mass and anthropometric and metabolic parameters in obese and overweight individuals. Methods: The study included one hundred people (85 women) with excess fat mass who voluntarily declared their willingness to participate in the weight-loss program consisted of 12 weeks of the DASH diet combined with regular, supervised physical activity. Anthropometric measurements and laboratory tests were performed in all study participants, and anthropometric and metabolic indices were calculated at baseline and after three months of treatment. Results: Body weight decreased significantly by 5.63 ± 4.03 kg, BMI by 2.06 ± 1.44 kg/m^2^, waist circumference by 5.6 ± 3.7 cm, fat mass from 40.04 ± 6.90 to 36.56 ± 7.07% and uric acid level by 16.0 ± 41.6 μmol/L (*p* < 0.001 in all cases). We also found an improvement in lipid profile and anthropometric and metabolic indices, except for HDL cholesterol and plasma glucose levels. Conclusions: The effect of the DASH diet and supervised physical activity was beneficial regardless of age, sex and the presence of hypertension or dysglycemia at baseline. The implementation of a healthy lifestyle was associated with a significant improvement in anthropometric and metabolic parameters, which, if continued, may reduce the risk of unfavorable health-related outcomes in the future.

## 1. Introduction

The prevalence of obesity has reached alarming rates in both industrialized and developing countries, becoming a serious public health issue. The problem is growing worldwide, reaching pandemic proportions. Globally, the number of people with obesity has increased almost threefold between 1975 and 2016, and currently, worldwide, overweight and obesity are responsible for more deaths than underweight [1]. According to data from the National Health Fund, the prevalence of obesity and overweight in Poland in 2019 was 55.6% [2]

Obesity adversely affects physiological functions of the body and poses a serious threat to public health. Excess body weight is associated with an increased risk of diabetes, cardiovascular diseases, malignant neoplasms and various musculoskeletal disorders. Moreover, obesity is associated with several neuropsychological disorders, including depression and mood alterations. Obesity also has a negative impact on quality of life, work efficiency and costs of medical care [3,4,5]. Eating and lack of exercise are two of the most important contributors to the development of obesity and its negative health consequences [3,4].

Hypertension is common among people with overweight and obesity, and its prevalence is almost two- and threefold higher, respectively, compared to that of the population with normal body weight [6]. An appropriate diet and physical activity leading to weight loss are the most important tools in the behavioral treatment of hypertension [7]. One of the dietary patterns dedicated to the treatment and prevention of hypertension is the DASH (Dietary Approaches to Stop Hypertension) diet, which is rich in fruits, vegetables and low-fat dairy products and reduced in saturated and total fat [8]. Regular physical activity is also associated with many benefits. It reduces fat mass, including intra-abdominal fat, and increases muscle and bone mass. Moreover, it lowers blood pressure and improves glucose metabolism, lipid profile and body performance [7].

Diabetes is also frequently associated with increased body weight. An observational cohort study conducted in a population of elderly women in central Poland revealed that 93.9% of females with diabetes had an excess of body weight, and the prevalence of overweight and obesity was 24.5% and 69.4%, respectively [9]. Therefore, Diabetes Poland strongly recommends behavioral interventions, including the DASH diet, as one of the methods for reducing excess body weight [10].

People with excess body weight consume significantly more healthcare budget resources than people with a healthy weight. In the member countries of the Organization for Economic Co-operation and Development (OECD), obesity and overweight account for 71% of all expenses for diabetes, 23% of the costs of treating cardiovascular diseases and 9% of the costs of treating cancer [11]. Pharmacotherapy for obesity, as well as bariatric surgery, are expensive on both individual and healthcare budget levels. Therefore, nonpharmacological interventions seem to be an attractive alternative to pharmacological and surgical therapy for obesity and its comorbidities [12].

The DASH diet, rich in fruits, vegetables and low-fat dairy foods and with reduced saturated and total fat, was developed for the treatment and prevention of hypertension. In the original study, both systolic as well as diastolic blood pressure (SBP and DBP) decreased significantly in subjects following this diet, which was more pronounced in hypertensive patients. However, its effect on body weight was rather small (0.4 kg) [8]. According to the position statement released in 2023, the DASH diet can be recommended for a reduction in body weight and waist circumference (WC) and the improvement of blood pressure, lipid profile, CRP (C-reactive protein) level and glycemic control (recommendation grade B) [12]. In the past, a number of studies were conducted to assess the effect of the DASH diet on blood pressure and some anthropometric and metabolic parameters. Their results were summarized in two meta-analyses. In the first one, including 13 studies, the DASH diet was effective in reducing body weight, BMI and WC [13]. The second meta-analysis assessed the effect of the DASH diet on blood pressure, glucose and blood lipids. Twenty randomized controlled trials were included in the analysis and showed a positive effect on blood pressure, total cholesterol and LDL cholesterol, while plasma glucose, HDL cholesterol and triglycerides did not change significantly [14].

Apart from diet, physical activity is also considered a cornerstone of nonpharmacological treatment for overweight and obesity [15]. However, the number of studies assessing the additive effect of the DASH diet combined with physical activity is limited. Moreover, participants usually receive only verbal advice to increase the number of steps per day, as measured by a pedometer, and only two studies included supervised exercise (cycling and/or walking/jogging) as part of the intervention, which is a clear evidence gap [16,17,18,19,20]. Therefore, we decided to combine the calorie-restricted DASH diet with regular physical activity in obese and overweight individuals to assess the synergistic effect of these interventions on body weight and various anthropometric and metabolic parameters and indices.

The aim of this study was to examine whether the DASH diet combined with tailored, supervised physical activity had a beneficial effect on body weight, fat mass, lipid profile, plasma glucose, uric acid levels and calculated anthropometric and metabolic indices in overweight and obese individuals participating in a weight-loss program conducted in a real-life setting. We also analyzed the effectiveness of this behavioral intervention in subgroups stratified by age, sex and presence of hypertension or dysglycemia to see if the intervention was equally effective in such subgroups.

## 2. Materials and Methods

### 2.1. Study Population

In the year 2016, the Medyk Healthcare Center in Rzeszów, the largest private healthcare institution in the region (the first author worked there at that time), announced through posters and flyers in all of its clinics information about the possibility of participating in a behavioral intervention program enabling body weight loss. People with overweight (BMI 25.0–29.9 kg/m^2^) or obesity (BMI ≥ 30 kg/m^2^) [1] willing to participate in a 12-week course of the DASH diet combined with an aerobic exercise program could report to the reception desk of the designated clinic in which the program was conducted, where they received detailed information about the program and an informed consent form to sign.

We aimed to recruit a group of 100 participants. Such a number was not based on an a priori power analysis but on an assumption that it would be sufficient for the analysis. Participants were eligible for the program if they were willing to be enrolled in the program and if they had been qualified by a general practice (GP) physician. Review of the patients’ medical records provided information about their baseline comorbidities: hypertension, dyslipidemia, dysglycemia (prediabetes or diabetes), thyroid disorders, cardiovascular disease, depression and gastro-esophageal reflux disease. These data were used to check for eligibility for participation in the study. The qualification or exclusion of participants in the program was at the discretion of the GP, who determined who was not suitable for vigorous physical activity, e.g., individuals with uncontrolled diabetes, hypertension, hypo- or hyperthyroidism, symptomatic cardiovascular disease, heart failure, asthma, chronic obstructive pulmonary disease, chronic kidney disease and other diseases making them incapable of intense physical activity. Blood pressure was measured using a standardized electronic blood pressure monitor, and hypertension was diagnosed according to the criteria of the European Society of Hypertension (ESH) and the European Society of Cardiology (ESC) (SBP ≥ 140 mm Hg and/or DBP ≥ 90 mm Hg [21]. All participants in the program were under the care a dietitian, a physiotherapist and a physician throughout the duration of the program.

A total of 123 people expressed a willingness to participate in the weight-loss program. Of these, sixteen were excluded by GP because of contraindications to vigorous physical activity, and seven (solely females) had a normal BMI. Finally, 100 people (85 women) who entered and completed the program were included in the analysis (Figure 1).

### 2.2. Assessments

Each study participant was supervised by a physician, dietitian and physiotherapist throughout the study. Body weight, height and fat mass were measured using the InBody120 analyzer (Biospace Co., Ltd., Seoul, Republic of Korea). WC was measured midway between the lower costal margin and the iliac crest, and hip circumference (HC) was assessed at the greatest protrusion of the buttocks using a tape measure according to standards described in the reference manual [22]. Based on these parameters, anthropometric indices were calculated:$$\mathrm{BMI}\;(\mathrm{body}\;\mathrm{mass}\;\mathrm{index})=\frac{weight\;(kg)}{height\;{\left(m\right)}^{2}}$$
$$\mathrm{WHR}\;(\mathrm{waist}\text{-}\mathrm{to}\text{-}\mathrm{hip}\;\mathrm{ratio})=\frac{WC\;(cm)}{HC\;(cm)}$$
$$\mathrm{WHtR}\;(\mathrm{waist}\text{-}\mathrm{to}\text{-}\mathrm{height}\;\mathrm{ratio})=\frac{WC\;(cm)}{height\;(cm)}$$

Blood samples for laboratory tests—cholesterol (total and HDL), triglycerides, plasma glucose and uric acid level—were collected during fasting and measured in the Medyk Healthcare Center laboratory using a Vitros 4600 analyzer (Cardinal Health, Dublin, OH, USA). LDL cholesterol was calculated using the Friedewald formula. Based on laboratory and anthropometric measurements, the following metabolic indices were calculated:

TG/HDL-C (triglycerides/HDL cholesterol ratio):$$\mathrm{TG}/\mathrm{HDL}\text{-}C=\frac{TG\;(mg/dL)}{HDL\text{-}C\;(mg/dL)}$$

TyG (triglycerides/glucose index):$$\mathrm{TyG}=\frac{\ln TG\left(mg/dL\right)\times glucose\;(mg/dL)}{2}$$

VAI (visceral adiposity index) was calculated according to the equation developed by Amato et al. [23].
$$\mathrm{VAI}\;(\mathrm{women})=\left[\frac{WC\;(cm)}{36.58+(1.89\times BMI)}\right]\times \left[\frac{TG\;(mg/dL)}{0.81}\right]\times \left[\frac{1.52}{HDL\text{-}C\;(mg/dL)}\right]$$
$$\mathrm{VAI}\;(\mathrm{men})=\left[\frac{WC\;(cm)}{39.68+(1.88\times BMI)}\right]\times \left[\frac{TG\;(mg/dL)}{1.03}\right]\times \left[\frac{1.31}{HDL\text{-}C\;(mg/dL)}\right]$$

### 2.3. Intervention

#### 2.3.1. Diet

Following a consultation with the dietitian, the participants received the meal plan for the duration of the program. Calorie intake was individually determined for each participant based on basal metabolic rate (BMR) and physical activity level (PAL), which were used to determine total metabolic rate (TMR). BMR was determined using the Harris–Benedict equation [24]:Women:BMR=655+9.5 (weight,kg)+1.9 (height,cm)−4.7 (age,years) kcal/day
Men:BMR=66+13.8 (weight,kg)+5.0 (height,cm)−6.8 (age,years) kcal/day

PAL was determined for study participants according to WHO recommendations: for people with a sedentary lifestyle, PAL = 1.40; for those with limited physical activity, PAL = 1.55–1.60; and for physically active persons, PAL = 1.75 [18]. TMR was calculated based on the following equation [25]:TMR=BMR×PAL kcal/day

The diet was designed in accordance with the DASH diet rules and its main assumptions: 5 servings of vegetables and fruit per day, about 7 servings of carbohydrates per day, 2 servings of low-fat dairy products per day, 2 or fewer servings of lean meat products per day and nuts and seeds 2–3 times per week [8]. Negative energy balance was applied individually with a deficit ranging from 500 to 1000 calories according to patients’ individual requirements calculated based on their TMR. The diet was designed to ensure that calorie intake exceeded the participant’s BMR in order to minimize the loss of lean body mass. Adherence to the diet was assessed monthly by interviewing the participants about the number of meals and their composition in the past week.

#### 2.3.2. Physical Activity

Each study participant completed a 12-week program of aerobic exercise. Each training session included aerobic and fitness exercises and spinning (dynamic riding on a special stationary spinning bike to the rhythm of music with an instructor). Each session comprised 10 min periods of warm-up, relaxation and stretching. The exercise program was carried out in the Medyk Healthcare Center under the supervision of a physiotherapist and consisted of two 60 min sessions per week of moderate intensity, corresponding to 50–70% of heart rate reserve (HRR). HRR was calculated according to the following formula [26]:$$\mathrm{HRR}=Max\;HR\left(220-age,years\right)-resting\;heart\;rate$$

### 2.4. Statistical Analysis

Statistical analysis was performed using SigmaPlot for Windows software, version 12.5 (Systat Software Inc., San Jose, CA, USA). The continuous variables are presented as mean and standard deviation (SD). The nominal variables are presented as absolute and percentage frequencies. The normality of data distribution was checked using the Shapiro–Wilk test. The differences between baseline and end-of-study values were analyzed using a paired two-tailed Student’s *t*-test for dependent variables. Differences between subgroups at baseline and at the end of the study according to age (above and below median), sex and presence of hypertension or dysglycemia were analyzed by repeated measures ANOVA. Heterogeneity between effects in these subgroups was analyzed using PQStat software version 1.8.6 (PQStat Software, Poznań, Poland). The categorical data were compared using the χ^2^ test. A *p* value <0.05 was considered significant.

## 3. Results

### 3.1. Whole Group

The mean age of study participants was 48.8 ± 12.1 years (range 19–75 years), weight 92.2 ± 16,8 kg, height 165 ± 8 cm and mean BMI 33.7 ± 4.9 kg/m^2^. At baseline, obesity was prevalent in 77 subjects, and hypertension was present in 50 individuals, while diabetes and prediabetes were recorded in 14 and 15 subjects, respectively (overall, 29 participants had dysglycemia). The baseline characteristics of study participants are presented in Table 1.

All enrolled subjects completed the program. Mean changes in analyzed parameters are summarized in Table 2. At the end of the study, mean body weight decreased by 5.63 ± 4.03 kg (*p* < 0.001). Overall, 96 participants lost weight by a mean of 6.0% compared to baseline. In 14 people, weight loss exceeded 10% (maximum 22.3%), in 47 cases it was 5–10% and in 35 persons it was <5%. Mean BMI decreased by 2.06 ± 1.44 kg/m^2^ (*p* < 0.001). At baseline, 77 subjects were obese, 22 were overweight and 1 had high normal weight (near 25 kg/m^2^). After 3 months of the calorie-restricted DASH diet combined with regular physical activity, the number of subjects with obesity decreased to 55 persons, and 4 subjects achieved normal body weight (*p* = 0.004) (1 person had high normal weight at baseline). WC at baseline in all but one study participants fulfilled criteria defined for metabolic syndrome (WC ≥ 80 cm in women and ≥94 cm in men) [27], and although it significantly decreased (*p* < 0.001) in almost all study participants, it normalized in two subjects only. Overall metabolic syndrome at baseline was prevalent in 27 women (31.8%) and nine men (60.0%). Mean fat mass at the end of the study remained elevated, but it highly significantly decreased compared to baseline (*p* < 0.001). Lipid profile and uric acid level also significantly improved, with the exception of HDL cholesterol concentration, which slightly but significantly decreased by 0.08 ± 0.23 mmol/L (*p* = 0.002). Triglycerides at baseline were elevated in 28 subjects and remained elevated in 17 subjects at the study end, but this difference was not significant (*p* = 0.090). Interestingly, fasting plasma glucose was not significantly different compared to baseline. A significant improvement was observed in all anthropometric (WHtR and WHR) and metabolic (TG/HDL-C, TyG and VAI) indices.

### 3.2. Subgroup Analysis

We divided participants into subgroups according to age (below and ≥48 years), sex and presence of hypertension or dysglycemia. Higher age was associated with a higher prevalence of hypertension and higher fat mass. People with hypertension were significantly older; they had also a higher BMI, WC, WHtR and WHR. The same results were observed in patients with dysglycemia. In addition, these participants also had a higher fat mass, fasting plasma glucose, uric acid level and TyG. Despite these significant differences between subgroups at baseline and at the end of the study, the beneficial effect of the intervention was homogenous in each subgroup regardless of age, sex and presence of hypertension or dysglycemia. Detailed results are presented in Appendix A.

In the linear mixed model analysis of covariates, weight change was the only factor independently associated with an improvement in metabolic biomarkers (total and LDL cholesterol, triglycerides and uric acid—there was no improvement in plasma glucose or HDL cholesterol levels), including after adjustment for age, sex and baseline prevalence of hypertension or dysglycemia. The same applied to metabolic indices.

## 4. Discussion

The DASH diet was developed as a tool for the dietary treatment of elevated blood pressure, and the original study did not analyze effect of this diet on other cardiovascular disease (CVD) risk factors [8]. Such studies and analyses have been conducted in later years. Soltani et al. conducted a meta-analysis of the DASH diet’s effect on body weight, BMI and WC and revealed a greater reduction in these parameters compared with other typical weight-loss diets [13]. In our study, after only three months of the calorie-restricted DASH diet combined with supervised aerobic exercise, mean body weight decreased by over 5.5 kg, BMI significantly improved and the number of obese patients decreased from 77 to 55 subjects. We also observed an improvement in other anthropometric parameters—WC and hip circumference—and indices—WHtR and WHR (in the latter case, although significant, the effect was very small due to a reduction in both WC and hip circumference). It is worth noting that WHtR is considered a better predictor of type 2 diabetes development compared to other anthropometric parameters [28,29] and is probably a better indicator than BMI for assessing cardiovascular risk in people with diabetes [30].

The effect of the DASH diet on CVD risk factors, including blood pressure, glucose level and lipid profile, was summarized in a meta-analysis by Siervo et al. The authors found a significant reduction in SBP, DBP and total and LDL cholesterol levels. No differences were found in glucose, HDL cholesterol or triglyceride levels [14]. At the end of our study, we observed a significant improvement in total and LDL cholesterol and in triglyceride concentrations, while HDL cholesterol levels significantly decreased and plasma glucose concentrations did not change. The latter may be due to the relatively high carbohydrate content in the DASH diet [8]. The total, LDL and HDL cholesterol level reduction was mainly due to the low fat content in the diet. Very similar findings were reported by Badali et al. in a study conducted in a population of patients with steatotic liver disease. Similar to our observations, they found a significant reduction in triglycerides, total and LDL cholesterol levels as well as the TG/HDL-C ratio [31]. A reduction in HDL cholesterol concentration can be considered an unfavorable effect, because a low level (which is frequently additionally associated with elevated levels of triglyceride-rich lipoproteins) is associated with elevated risk of cardiovascular events and all-cause mortality [32]. Apart from the negative impact on HDL cholesterol, we observed several beneficial effects of the treatment. Among them, it is worth emphasizing the reduction in uric acid level. Elevated serum uric acid level is considered a leading cause of gout. Moreover, there is also a relationship between elevated uric acid level and hypertension, obesity, dyslipidemia and cardiovascular disease (CVD) [33]. In our study, uric acid level was significantly reduced, which is in line with a recently published meta-analysis [34].

Also, all three metabolic indices analyzed in our study significantly improved during observation. The TG/HDL-C ratio is a strong predictor of type 2 diabetes development in the elderly female population [27] and can also be considered a marker of CVD risk, as well as all-cause and cardiovascular death [35,36]. A systematic review and meta-analysis published in 2022 revealed an association between an elevated TyG index and coronary artery disease (CAD), myocardial infarction (MI) and overall CVD, but no association was found between TyG index and cardiovascular death or all-cause mortality [37]. Furthermore, data from large cohort studies indicate relationship between TyG index and the risk of heart failure [38]. The VAI is considered a useful tool in the assessment of overall cardiometabolic risk. It is significantly correlated with a history of myocardial infarction, higher carotid intima-media thickness, diabetes, prediabetes and impaired kidney function in women aged 65–74 years [39]. It is also an independent predictor of the development of type 2 diabetes [29] and cardiovascular and all-cause mortality [39,40]. Therefore, a significant reduction in these three metabolic indices may be beneficial in the future by reducing the risk of unfavorable health-related outcomes.

The design of all studies discussed above did not include the addition of physical activity to the DASH diet. In fact, the number of studies examining the effects of the DASH diet in combination with any physical activity is not large. The ENCORE study, with 144 participants, revealed a beneficial effect of adding exercise and weight loss to the DASH diet on blood pressure, body composition, glucose and insulin levels, insulin sensitivity and lipid parameters, as well as vascular and autonomic function, with a reduction in left ventricular mass [16,17]. Another randomized trial comprising 40 participants also confirmed a better effect of a 12-week exercise plus DASH diet intervention on blood pressure and autonomic nervous system function compared with exercise alone [18]. A study conducted in 40 patients with uncontrolled hypertension revealed a significantly better impact on blood pressure of the DASH diet associated with advice to increase walking using a pedometer compared to that in participants following a diet based on the American Diabetes Association recommendations [19]. Another study that analyzed a wide range of anthropometric and metabolic parameters was conducted exclusively in a diabetic population. This randomized study comprising 35 participants with type 2 diabetes compared two groups of patients: one group received only DASH dietary guidance, whereas the other group (DASHPED) received dietary guidance with the recommendation to walk with a pedometer. Both groups showed significant reductions in body weight and BMI. However, no differences between groups were found in these and other analyzed parameters (muscle mass, body fat, waist/hip ratio, glycemic control, lipid profile and insulin sensitivity) [20]. Considering the relatively small number of studies evaluating the combined effect of the DASH diet and physical activity on anthropometric and metabolic parameters in people with excess body weight, our work and the design of our study can be considered novel because in our weight-loss program, the use of the DASH diet was combined with tailored, systematic, supervised aerobic exercise. Moreover, our analysis was extended by the analysis of the effect of this behavioral intervention among different subgroups of participants according to their age, gender and the presence of hypertension or glucose metabolism disorders.

Obviously, our study is not free from several limitations. The first is the relatively small sample size, which affected the statistical power of our findings. The second limitation is the retrospective design; thus, some important data were not available, e.g., blood pressure measurements or HbA1c level in participants with diabetes. Another limitation is the lack of a control group; thus, no randomization of study participants was performed. We also did not use standardized protocols for the assessment of adherence to dietary recommendations, which makes this assessment less reliable. The last limitation is a relatively short follow-up period. Nevertheless, our study has also some strengths: we analyzed a wide range of anthropometric, laboratory and metabolic parameters; also, body composition was assessed at baseline and at the end of the weight-reduction program. Our results indicate that the DASH intervention with regular physical activity had beneficial effects on improving anthropometric and metabolic profiles in our study participants, regardless of sex, age and the presence of hypertension or dysglycemia. It is obvious that further, longer-term, prospective, randomized controlled trials are needed to confirm the beneficial effects of behavioral therapy based on the DASH diet and regular physical activity, as well as to assess long-term cardiovascular and other benefits of such treatment.

## 5. Conclusions

To conclude, the DASH diet remains among the most recommended diets by scientific organizations for achieving and maintaining metabolic and cardiovascular health. When combined with regular physical activity, it significantly improved anthropometric and metabolic profiles in our study participants, and the benefits were evident regardless of age, sex and the presence of hypertension or dysglycemia. Long-term adherence to such behavioral changes may be beneficial for health outcomes and can be widely recommended to almost all overweight and obese individuals, who currently constitute more than 50% of the adult population in OECD countries. Therefore, it seems highly justified to increase public awareness of the role of behavioral changes in reducing the prevalence of obesity and improving health-related outcomes at the population level.

## Figures and Tables

**Figure 1 nutrients-16-03187-f001:**
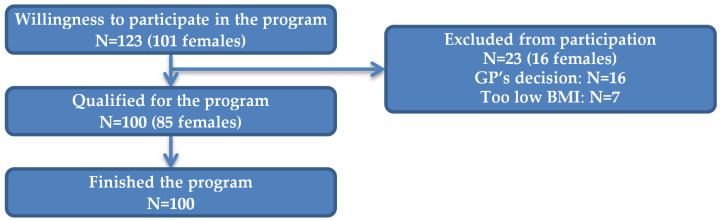
Flowchart of the study participants.

**Table 1 nutrients-16-03187-t001:** Baseline characteristics of the study population. Significant differences between men and women are marked in ***italic bold***.

Parameter	Women, n = 85	Men, n = 15	*p* Value
**Age range (years), n (%):** <45 45–54 ≥55	32 (37.6%) 22 (25.9%) 31 (36.5%)	7 (46.7%) 4 (26.7%) 4 (26.7%)	0.705
**BMI category, n (%):** Normal weight/overweight Obesity I grade Obesity II and III grade	23 (27.1%) 37 (43.5%) 25 (29.4%)	6 (40.0%) 9 (60.0%)	** *0.022* **
**Hypertension, n (%):** Yes No	41 (48.2%) 44 (51.8%)	9 (60.0%) 6 (40.0%)	0.575
**Glucose metabolism, n (%)** Diabetes Prediabetes Normoglycemia	12 (14.1%) 9 (10.6%) 64 (75.3%)	2 (13.3%) 6 (40.0%) 7 (46.7%)	** *0.012* **

All analyses were performed using χ^2^ statistics.

**Table 2 nutrients-16-03187-t002:** Anthropometric and metabolic parameters and indices at baseline and at the end of the study. Significant differences are marked in ***italic bold***.

Parameter	Baseline Mean ± SD or n (%)	End of Study Mean ± SD or n (%)	Difference Mean ± SD	*p* Value
Weight (kg)	92.2 ± 16.8	86.5 ± 15.8	−5.6 ± 4.0	*<0.001*
BMI (kg/m^2^)	33.7 ± 4.9	31.7 ± 4.6	−2.1 ± 1.4	*<0.001*
Obesity II and III grade, n (%) Obesity I grade, n (%) Overweight, n (%) Normal weight, n (%)	34 (34%) 43 (43%) 22 (22%) 1 (1%)	19 (19%) 36 (36%) 39 (39%) 5 (5%)	-	*0.004* *
Waist (cm)	102.6 ± 13.8	97.0 ± 13.6	−5.6 ± 3.7	*<0.001*
WHtR	0.62 ± 0.08	0.59 ± 0.08	−0.03 ± 0.02	*<0.001*
Hip (cm)	115.5 ± 9.1	110.1 ± 8.9	−5.4 ± 3.9	*<0.001*
WHR	0.89 ± 0.09	0.88 ± 0.10	−0.01 ± 0.04	*0.033* ^†^
Fat mass (%)	40.0 ± 6.9	36.6 ± 7.1	−3.5 ± 2.8	*<0.001*
Total cholesterol (mmol/L)	5.37 ± 1.29	4.89 ± 0.89	−0.49 ± 0.74	*<0.001*
Triglycerides (mmol/L)	1.42 ± 0.58	1.24 ± 0.52	−0.18 ± 0.48	*<0.001*
HDL cholesterol (mmol/L)	1.49 ± 0.40	1.41 ± 0.34	−0.08 ± 0.23	*0.002*
LDL cholesterol (mmol/L)	3.27 ± 0.98	2.90 ± 0.83	−0.36 ± 0.68	*<0.001*
Glucose (mmol/L)	5.37 ± 0.95	5.33 ± 0.92	−0.04 ± 0.73	0.466
Uric acid (μmol/L)	316.0 ± 78.0	300.0 ± 73.3	−16.0 ± 41.6	*<0.001*
TG/HDL-C ratio	2.48 ± 1.59	2.21 ± 1.26	−0.27 ± 1.04	*0.030* ^†^
TyG index	4.66 ± 0.22	4.59 ± 0.22	−0.07 ± 0.16	*<0.001*
VAI	4.35 ± 2.52	3.82 ± 1.88	−0.53 ± 1.78	*0.010*

SD: standard deviation; BMI: body mass index; WHtR: waist-to-height ratio; WHR: waist/hip ratio; TG/HDL-C: triglycerides/HDL cholesterol ratio; TyG: triglycerides/glucose index; VAI: visceral adiposity index. All statistical analyses were performed using paired, two-tailed Student’s *t*-test for dependent variables, except * calculated by χ^2^. **^†^** statistical power <0.8.

## Data Availability

The dataset for this study is available at the University of Rzeszów Repository.

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
