# Peer review of "Beneficial Effect of Dietary Approaches to Stop Hypertension Diet Combined with Regular Physical Activity on Fat Mass and Anthropometric and Metabolic Parameters in People with Overweight and Obesity"

_nutrients, 2024, doi:10.3390/nu16183187_

Round 1

Reviewer 1 Report

Comments and Suggestions for Authors

Dear authors, thank you for the opportunity to review this manuscript. This is a pre-and post-intervention study, assessing the effects of a 12-week combined DASH diet and physical activity on the changes in anthropometric and metabolic parameters in overweight and obese adults in Rzeszów, and in subgroups. The major concerns of this study include the flaws in study design, data measurement/assessment and statistical analysis. In particular, this is pre- and post-intervention study without comparison with control group, thus we can't reliably attribute the effects on the outcomes to the intervention, if other factors are present. In addition, there is a lack of novelty. This study has been conducted in many other settings showing the associations between DASH diets and obesity-related anthropometric parameters and metabolic biomarkers.

There are other issues which need to be addressed carefully, listed as below. I hope these thoughts help as you progress this work.

Introduction:

Suggest the authors to describe the degree and extent of the prevalence of overweight and obesity, or/and hypertension and metabolic syndrome, in the study population.

·       The objective needs to be clearer. In the subgroup analysis, apart from hypertension, the authors should also include other subgroups being analysed such as age, sex, and in the presence dysglycemia.

Abstract:

·       Please describe the study design, setting (country) or the characteristics of the participants recruited.

·       Conclusion (line 23-24) does not accurately reflect the implications of this study, please revise accordingly.

Methodology:

·       Inclusion criteria for participant selection needs further clarification. Any restriction such as on age and health condition? In line 74-76, the participants’ health conditions or medical records, however no description on how this information was used.

·       The bioanalytical method used to measure blood biomarkers e.g., TG, HDL-C, glucose etc. should be described. Please mention how blood pressure was measured and how hypertension was defined.

·       There is no description on dietary assessment. How baseline and post-intervention dietary intake has been assessed? Moreover, how was the compliance or adherence to the prescribed DASH diet and physical activity being measured? This needs to be described.

·       The duration of the intervention (12-week) should me mentioned as the whole intervention (in line 67?), rather at line 114 only for physical activity.

·       Appropriate statistical analysis should be used to analyse the subgroup differences in response to the intervention - repeated Measures ANOVA rather than unpaired two-tailed Student’s t-test or by Mann-Whitney rank sum test (line 126-129)

Results:

·       When reporting categorical variables (Table 1, as well as in text), please report percentage alongside with number.  

·       The dietary intake at baseline and post-intervention should be reported.

·       The baseline general characteristics of the study participants by subgroups (e.g., sex, age, hypertension) should be reported as Table 1. The current Table 1 perhaps can be labelled as Table 2.

Discussion:

·       The discussion is rather superficial, and there is no further discussion on the subgroup analyses.  

Comments on the Quality of English Language

English level is acceptable, some grammar checking/editing and proofreading is needed. 

Author Response

Reply to Reviewer 1.

In my first word let me thank you for you valuable and inspiring comments, remarks, suggestions and recommendations. Thank to them we substantially rebuild our manuscript, which, hopefully, also significantly improved its quality. Please find the detailed responses below and you will find all the revisions and corrections highlighted in the re-submitted files.

Comment 1. Dear authors, thank you for the opportunity to review this manuscript. This is a pre-and post-intervention study, assessing the effects of a 12-week combined DASH diet and physical activity on the changes in anthropometric and metabolic parameters in overweight and obese adults in Rzeszów, and in subgroups. The major concerns of this study include the flaws in study design, data measurement/assessment and statistical analysis. In particular, this is pre- and post-intervention study without comparison with control group, thus we can't reliably attribute the effects on the outcomes to the intervention, if other factors are present. In addition, there is a lack of novelty. This study has been conducted in many other settings showing the associations between DASH diets and obesity-related anthropometric parameters and metabolic biomarkers.

Response 1. Thank you for this remark. We agree that lack of control group is a weak side of this study. Unfortunately we had limited (own) resources to conduct this study and adding additional participants would entail too high costs. Regarding novelty, we do not fully agree with this statement, because a number of studies analyzing effect of DASH diet combined with tailored, supervised aerobic exercise is rather scarce (usually, participants in other studies were given a pedometer and recommendations to increase their daily step count). Also, in many, especially earlier studies, metabolic biomarkers were not analyzed.

There are other issues which need to be addressed carefully, listed as below. I hope these thoughts help as you progress this work.

Introduction:

Comment 2. Suggest the authors to describe the degree and extent of the prevalence of overweight and obesity, or/and hypertension and metabolic syndrome, in the study population.

Response 2. Thank you for this suggestions – done. The study population is more precisely described in Material and Methods Chapter

Comment 3. The objective needs to be clearer. In the subgroup analysis, apart from hypertension, the authors should also include other subgroups being analysed such as age, sex, and in the presence dysglycemia.

Response 3. Thank you for this recommendation. We added this information at the end of introduction.

Abstract:

Comment 4. Please describe the study design, setting (country) or the characteristics of the participants recruited.

Response 4. Thank you for pointing it out. Done

Comment 5. Conclusion (line 23-24) does not accurately reflect the implications of this study, please revise accordingly.

Response 5. Thank you for this remark. We fully agree – we removed it.

Methodology:

Comment 6. Inclusion criteria for participant selection needs further clarification. Any restriction such as on age and health condition? In line 74-76, the participants’ health conditions or medical records, however no description on how this information was used.

Response 6. Thank you for this remark. We corrected these data accordingly and described how these data were used.

Comment 7. The bioanalytical method used to measure blood biomarkers e.g., TG, HDL-C, glucose etc. should be described. Please mention how blood pressure was measured and how hypertension was defined.

Response 7. We also thank for this remark and recommendation. Done.

Comment 8. There is no description on dietary assessment. How baseline and post-intervention dietary intake has been assessed? Moreover, how was the compliance or adherence to the prescribed DASH diet and physical activity being measured? This needs to be described.

Response 8. Thank you for this remark. We extended description and gave more details of dietary part of intervention under point 2.3.1.

Comment 9. The duration of the intervention (12-week) should me mentioned as the whole intervention (in line 67?), rather at line 114 only for physical activity.

Response 9. Thank you for this suggestion. We described it in Materials and Methods paragraph.

Comment 10. Appropriate statistical analysis should be used to analyze the subgroup differences in response to the intervention – repeated Measures ANOVA rather than unpaired two-tailed Student’s t-test or by Mann-Whitney rank sum test (line 126-129)

Response 10. Thank you for this remark and recommendation. I fully agree – my fault. We performed analysis according to your suggestion. However, according to Reviewer 3 remarks and recommendation to eliminate subgroup analysis, we decided to not fully eliminate it, but submit the tables with subgroups analysis as a supplementary material

Results:

Comment 11. When reporting categorical variables (Table 1, as well as in text), please report percentage alongside with number.  

Response 11. Thank you for this recommendation. Done

Comment 12. The dietary intake at baseline and post-intervention should be reported.

Response 12. As I wrote above, we assessed adherence to DASH diet only by taking a history from participants (the obtained results, at least in part, support the thesis of good adherence – perhaps the motivation to lose weight also played a role)

Comment 13. The baseline general characteristics of the study participants by subgroups (e.g., sex, age, hypertension) should be reported as Table 1. The current Table 1 perhaps can be labelled as Table 2.

Response 13. Thank you for this suggestion - we fully agree. Done

Discussion:

Comment 14. The discussion is rather superficial, and there is no further discussion on the subgroup analyses.  

Response 14. Thank you for this remark - you are absolutely right – the discussion was too poor and superficial. We corrected it and enriched in several new references and we are hoping that now it looks much better.

Once again we want to thank you for your valuable and inspiring comments, remarks, suggestions and recommendations which, hopefully, improved the quality of our manuscript.

Sincerely,

Mariusz DÄ…browski

Reviewer 2 Report

Comments and Suggestions for Authors

This is a clear and well-written article evaluating the effects of three months of Dash diet and physical activity in 100 overweight/obese Polish adults. The content is original, and the statistical analysis is correct.

My contribution to further improvement of the manuscript concerns the following aspect: since anthropometric measurements constitute an important part of the study, it is appropriate to give a few citations of anthropometry texts used as methodological reference (e.g., Lohman, T.G.; Roche, A.F.; Martorell, R. Anthropometric Standardization Reference Manual; Human Kinetics Books: Champaign, IL, USA, 1988). Also, for fairness, I suggest using the term standing height instead of the generic height. Finally, although the unit of stature is correctly denoted in cm, both in the text and Table 3,4,5 the mean value is given in meters. This is a serious error that absolutely must be corrected.

Minor concerns:

-       Lines 123-124: change “numbers and percentage” to “absolute and percentage frequencies”.

-       Line 131: move the 2 in the chi-square symbol to the superscript.

-       Table 1: indicate the statistical tests used for the different variables. For example, you can put a symbol next to the p-values with an explanation in the footnote.

Comments on the Quality of English Language

None

Author Response

Reply to Reviewer 2.

In my first word let me thank you very much on behalf of all the authors for your kind general comment and for the remarks regarding our manuscript. We followed all your suggestions and recommendations. Please find the detailed responses below and you will find all the revisions and corrections highlighted in the re-submitted files.

Reviewer’s 2 general comment

This is a clear and well-written article evaluating the effects of three months of Dash diet and physical activity in 100 overweight/obese Polish adults. The content is original, and the statistical analysis is correct.

My contribution to further improvement of the manuscript concerns the following aspect: since anthropometric measurements constitute an important part of the study, it is appropriate to give a few citations of anthropometry texts used as methodological reference (e.g., Lohman, T.G.; Roche, A.F.; Martorell, R. Anthropometric Standardization Reference Manual; Human Kinetics Books: Champaign, IL, USA, 1988). Also, for fairness, I suggest using the term standing height instead of the generic height. Finally, although the unit of stature is correctly denoted in cm, both in the text and Table 3,4,5 the mean value is given in meters. This is a serious error that absolutely must be corrected.

Response. Thank you for your kind opinion and presented remarks and suggestions. We agree that anthropometric measurements are important part of this study and we included suggested citation to the reference list. We also, according to your suggestion, changed the simple term “height” to “standing height” and we corrected obvious mistake regarding units of height and changed the values into centimeters.

Minor concerns:

Comment 2. Lines 123-124: change “numbers and percentage” to “absolute and percentage frequencies”.

Response 2. Thank you for this suggestions – we followed your recommendation and we changed the phrase according to your request.

Comment 3. Line 131: move the 2 in the chi-square symbol to the superscript.

Response 3. Thank you for this remark and perceptiveness - we changed 2 to superscript.

Comment 4. Table 1: indicate the statistical tests used for the different variables. For example, you can put a symbol next to the p-values with an explanation in the footnote.

Response 4. Thank you for this suggestion. We followed your recommendation. However, according to Reviewer’s 1 recommendation we added a table with baseline description of study population, thus, former Table 1 became Table 2. Also, following Reviewer’s 3 suggestion, we removed tables 2-5 with results of subgroups analysis from the main text and we moved them to supplementary materials as Tables S1-S4.

In the end let me thank you once again for your vauble comments, remarks, suggestions and recommendations which - hopefully - improved quality of our manuscript

Sincerely,

Mariusz DÄ…browski

Reviewer 3 Report

Comments and Suggestions for Authors

The current study reports on the effect of the DASH diet plus physical activity on cardiometabolic outcomes. There are several major issues with this manuscript. 

1.      The Introduction does not really discuss the need for the current study. Alterations in dietary intake and increased physical activity are known to effect cardiometabolic outcomes.

2.      It is strongly suggested that a flowchart be included.

3.      It appears that there was no random selection, and therefore, this is a sample of convenience. If this is correct, it should be noted and its implications discussed.

4.      For clarity, 2.3 and 2.4 can be included under a heading of Intervention. right now the reader expects to read how diet and physical activity were assessed, not the prescription.

5.      Were there any assessments of adherence to the DASH diet or levels of physical activity?

6.      It is strongly suggested that a power analysis be included. The sample size of 100 seems arbitrary.

7.      There is no rationale provided in the Introduction for conducting age or sex analyses. Concerning the age, sex, hypertension, and glycemic sub-analyses, 1) they add numerous analyses inflating the Type I error rate, 2) the findings are quite similar within subgroups, and 3) there is very little discussion of these differences in the Discussion. In short, these analyses do not appear to be adding substantial content to the manuscript. For these reasons, it is recommended that are eliminated.

8.      The significance values for WC and fat mass changes should be provided in text.

9.      I suggest changing “insignificant” to “non-significant” or something similar. 

10.  The statement “A significant improvement was observed in all anthropometric and metabolic indices.” Needs revision, as glucose is a metabolic outcome. Perhaps “…all other…”

11.  There does not seem to be any control for Type I error rate.

12.  The first two paragraphs of the Discussion seems unnecessary. They do not provide a discussion of the results or their implications. The content seems more appropriate for the Introduction.

13.  The novelty of the study is not provided in the Discussion.

14.  Relatedly, there is not much discussion of how the current study fits in with previous research. A meta-analysis was noted, but there have been other studies of the DASH diet combined with physical activity.

Author Response

Reply to Reviewer 3.

First of all, I would like to thank you for your valuable and very insightful comments, remarks, suggestions and recommendations. Please find the detailed responses below and you will find all the revisions and corrections highlighted in the re-submitted files.

The current study reports on the effect of the DASH diet plus physical activity on cardiometabolic outcomes. There are several major issues with this manuscript. 

Comment 1. The Introduction does not really discuss the need for the current study. Alterations in dietary intake and increased physical activity are known to effect cardiometabolic outcomes.

Response 1. I am sorry, but I do not fully agree with the first statement. Obesity is becoming the greatest public health problem worldwide. It is expected, that the number of people with obesity will exceed one billion persons before the year 2030. Any approach to fight this problem is reasonable. We know that different dietary approaches are effective in decreasing weight, especially with increased physical activity. However, in majority of published studies assessing effectiveness of such approach (which were not so numerous), participants usually received pedometer and recommendation to increase the number of steps. In our study it was tailored to 50-70% of heart reserve rate of any individual participant, supervised by physiotherapist, aerobic exercise repeated twice a week (which is described in methodology) and that is the difference. We were expecting that such approach will be far more effective than simple physical activity advise. And testing this hypothesis was the aim of our study.

Comment 12. The first two paragraphs of the Discussion seems unnecessary. They do not provide a discussion of the results or their implications. The content seems more appropriate for the Introduction.

Response 12. Thank you for this comment. We fully agree with your statement and we moved these paragraphs to Introduction.

Comment 2. It is strongly suggested that a flowchart be included.

Response 2. Thank you for this suggestion. We added a flowchart of study participants as a Figure 1.

Comment 3. It appears that there was no random selection, and therefore, this is a sample of convenience. If this is correct, it should be noted and its implications discussed.

Response 3. Thank you for this comment. In fact, in the first version of manuscript it would look like there was no random selection. In fact it was. In 2016 the Medyk Healthcare Center in Rzeszów (where the first author worked at that time) announced the behavioral intervention program enabling body weight loss. The program was open for any obese or overweight person treated by general practice physician in that Healthcare Center, who voluntarily wanted to participate. As it was described in methodology, these volunteers had to be examined and qualified by physician and if they had contraindications to physical exercise of predefined intensity, they were disqualified from participation. We were aimed to collect 100 participants due to limited resources for the study, and we presumed that such number will be enough to reject the null hypothesis, because we expected differences large enough to prove effectiveness of this weight loss program. We extended this description in the revised main text.

Comment 4. For clarity, 2.3 and 2.4 can be included under a heading of Intervention. Right now the reader expects to read how diet and physical activity were assessed, not the prescription.

Comment 5. Were there any assessments of adherence to the DASH diet or levels of physical activity?

Response 4/5. Thank you for these comments and suggestions. We fully agree with the first suggestion and we included diet and physical activity as a subheadings under the heading “Intervention”. However, we do not fully agree with the latter part of Comment 4. – in my opinion it is also of importance how the meal plan was designed and how in each individual case a calorie intake was calculated for the program participants – we described it in more details. Adherence to dietary prescriptions was assessed by the history taken from the participant at every physical training visit (twice a week). Although we wrote in original version that physical activity was supervised, in the revised version we clearly stated that each physical training (two 60-min sessions a week) was supervised by physiotherapist.

Comment 6. It is strongly suggested that a power analysis be included. The sample size of 100 seems arbitrary.

Response 6. Thank you for this suggestion. It is of utmost importance. We performed power analysis for each calculation and we marked P values with power <0.8 with symbol “†” and noted it in the footer. Regarding sample, we described it in Response 3.

Comment 7. There is no rationale provided in the Introduction for conducting age or sex analyses. Concerning the age, sex, hypertension, and glycemic sub-analyses, 1) they add numerous analyses inflating the Type I error rate, 2) the findings are quite similar within subgroups, and 3) there is very little discussion of these differences in the Discussion. In short, these analyses do not appear to be adding substantial content to the manuscript. For these reasons, it is recommended that are eliminated.

Response 7. Thank you for this critical comment. We included the rationale of assessing these sub-analyses at the end of Introduction. I can also add, that such analyses of different subgroups are conducted in many other studies, not only in the field of nutrition and such sub-analyses are of value, because – in my opinion – it is important to assure which patients/subjects benefit from any kind/type of intervention, and which do not. We agree with opinion that statistical power of some analyzes is low and the risk of the Type I error exists. Hence, we performed once again calculation of statistical power of each analyzed parameter and all P values <0.05 with statistical power <0.800 we marked with “”. And they were not so numerous, therefore, we did not removed these data, but moved the tables to supplementary materials, leaving short description of results obtained in these subgroups in the main text under subheadings.

Comment 8. The significance values for WC and fat mass changes should be provided in text.

Response 8. Thank you for this recommendation. Done.

Comment 9. I suggest changing “insignificant” to “non-significant” or something similar. 

Response 9. Thank you for this remark. Done.

Comment 10. The statement “A significant improvement was observed in all anthropometric and metabolic indices.” Needs revision, as glucose is a metabolic outcome. Perhaps “…all other…”

Response 10. I agree that glucose is a metabolic outcome, but metabolic indices are: TG/HDL-C ratio, TyG and VAI, that is why we used this statement.

Comment 11. There does not seem to be any control for Type I error rate.

Response 11. We performed analysis of statistical power in the whole group and in all but 2 cases (WHR and TG/HDL-C ratio) it is equal or close to 1.000, only in those 2 cases it was <0.8 (in the latter case precisely 0.736)

Comment 13. The novelty of the study is not provided in the Discussion.

Response 13. Thank you for this comment. We included such statement and explanation.

Comment 14. Relatedly, there is not much discussion of how the current study fits in with previous research. A meta-analysis was noted, but there have been other studies of the DASH diet combined with physical activity.

Response 14. Thank you for this remark. We fully agree with this opinion and we extended discussion addressing our findings to several other studies and we added several new references.

At the end let me once again thank you for your impact, for many relevant, insightful and important comments, remarks, suggestions and recommendations. And although I perceived a tone of some of them as a little bit too offensive, I am very grateful for all your comments, because they allowed us to improve the quality of our manuscript (hopefully substantially).

Sincerely,

Mariusz DÄ…browski

Reviewer 4 Report

Comments and Suggestions for Authors

This eminent article shows the benefits of the popular DASH-diet in fighting obesity. The set-up and work-up are well done. I have some comments:

- There exists a remarkable imbalance between men and women. Is there an explanation for this?

- Tables 2-5: the men's subgroups are rather small; are statistics appropriate?

Comments on the Quality of English Language

Some errors concerning sentence structure.

Author Response

Reply to Reviewer 4.

First of all, I would like to thank you for your kind opinion and valuable remarks. Please find the detailed responses below and you will find all the revisions and corrections highlighted in the re-submitted files.

This eminent article shows the benefits of the popular DASH-diet in fighting obesity. The set-up and work-up are well done. I have some comments:

Comment 1. There exists a remarkable imbalance between men and women. Is there an explanation for this?

Response 1. Thank you for this remark. In the methodology section we described in more details recruitment process. And because participation was voluntary it documents that women have probably higher motivation to lose weight than men, which is also apparent in Table 1, which was added according to recommendation of one of the Reviewers (the men feel do need to fight the weight when they are really obese ;-) We added short comment regarding this in the Discussion.

Comment 2. Tables 2-5: the men's subgroups are rather small; are statistics appropriate?

Response 2. Thank you for this remark. One of the Reviewers had also such doubts, thus, we performed analysis of statistical power of all statistical analyzes, and those P values <0.05 with statistical power <0.800 we marked with symbol “”.

At the end let me once again thank you for your time, kind opinion and valuable remarks.

Sincerely,

Mariusz DÄ…browski

Round 2

Reviewer 1 Report

Comments and Suggestions for Authors

Thank you, the authors, for their efforts in revising the manuscript. Some of the issues have been addressed; however, my major concern remains with the statistical analysis approach. As we all known, the reduction in the metabolic biomarkers (e.g., lipid profile, glucose) may be due to the intervention itself or the weight loss induced by that intervention. The authors need to adjust for the influences of body weight changes induced by the intervention on the metabolic biomarkers, using body weight changes as a covariate in the linear mixed model. The model will also need to be adjusted by sex and age (or presence of hypertension and other metabolic syndrome, if necessary).

Other suggested changes are as follows:

Abstract

·       - Please mention that this is a pre-post study design in the abstract.

Methodology

·       - Line 109 – A reference for the definition of overweight and obesity (BMI>25) is needed.

·       - Line 119 – Not all the listed medical histories were used for statistical analysis. It might be better to say “used to check for eligibility”??

·       - Lines 125-126 - Equipment used to measure blood pressure, and the cutoffs applied to define hypertension should be mentioned.

·       - Additionally, please detail the bioindicators and cutoffs applied to define dyslipidaemia, and dysglycemia (prediabetes or diabetes), including the references used.

·       - Line 180 – The authors state that the adherence to the intervention was assessed by interviewing participants, but it is unclear  how these interviews were conducted, and what questions were asked? This should be clarified or, at the very least, addressed in the limitation section if there is no standardised protocol, including its potential impacts on the results. Same applies to the physical activity assessment.

Result

·       - Lines 240-270 - It is very uncommon for a large section of the results to be based on findings from supplementary materials. If the authors decide to report these findings, perhaps they could include one table in the main manuscript, e.g., report only the differences and p values for the subgroups.

Discussion

·       - Lines 333-335 – There appears to be an error in this sentence. I assume the authors mean to say: “…..None of these studies analyzed metabolic biomarkers other than anthropometric indices (e.g., body weight and BMI) and hypertension. Metabolic indicators such as lipid profiles, blood glucose, and uric acid levels were not assessed”. ?

·       - For Lines 335-336 – As far as I know, that is not the only study. For instance, the paper published by the same authors for reference [16], they assessed the impacts of the same interventions reported in [reference 16] (DASH diet alone and in combination with exercise and caloric restriction) on insulin sensitivity and lipids, in addition to blood pressure and cardiovascular markers in.

-          Blumenthal et al. (2010) Hypertension. 55(5): 1199–1205. doi: https://doi.org/10.1161/HYPERTENSIONAHA.109.149153   

Please kindly check and revise the wording, esp regarding terms such as ‘first study’ or ‘only study’.  

·       - I also recommend a more detailed discussion on the limitations of the study design (pre-post), including how this may have impacted the results.

Conclusion

·       - It is uncommon to have citations in the conclusion, please remove.  

·      -  Due to the pre-and-post study design, I would suggest rephrasing the sentence in Lines 367-371 as: “Our study indicates that the DASH intervention with regular physical activity has beneficial effects in improving anthropometric and metabolic profiles in our study participants, regardless of sex, age, or presence of hypertension or dysglycemia.” The current phrasing sound like if they have implied a comparison of DASH vs. DASH + physical activity or others (e.g., control).

Author Response

Reply to Reviewer 1

At first word let me thank you for this  insightful and constructive review with comments, remarks suggestions and recommendations. We tried to follow all of them. And we hope that results of our efforts will be satisfactory for you.

Reviewer’s 1 comments

General comment:

Thank you, the authors, for their efforts in revising the manuscript. Some of the issues have been addressed; however, my major concern remains with the statistical analysis approach. As we all known, the reduction in the metabolic biomarkers (e.g., lipid profile, glucose) may be due to the intervention itself or the weight loss induced by that intervention. The authors need to adjust for the influences of body weight changes induced by the intervention on the metabolic biomarkers, using body weight changes as a covariate in the linear mixed model. The model will also need to be adjusted by sex and age (or presence of hypertension and other metabolic syndrome, if necessary).

Response.

Thank you for your valuable and positive opinion regarding our efforts. In fact, we tried to address a number of issues. And I am aware that some issues regarding statistics may remain. But I am clinician and not statistician. I treat people with diabetes, who are frequently obese or extremely obese. And I know from a number of clinical studies that weight reduction itself is beneficial. In studies with anti-obesity medications it was beneficial for plasma glucose level, both for patients with and without diabetes (STEP Program with semaglutide), for the heart – it reduced incidence of  MACE (Major Cardiovascular Events: cardiovascular deaths and non-lethal myocardial infarctions and strokes) in SELECT Study, it reduced risk of hospitalizations for heart failure (STEP-HFpEF and STEP-HFpEF DM randomized trials), it reduced risk of kidney function deterioration and development of end-stage kidney disease (FLOW trial with semaglutide), weight reduction in SURPASS and SURMOUNT programs with tirzepatide was associated with highly significant reduction of steatosis and steato-hepatitis in the course of metabolism-dysfunction associated steatotic liver disease or steato-hepatitis (MASLD / MASH). We can discuss here whether these benefits were caused by medication and its action on many receptors disseminated throughout the body or by weight loss itself. I don’t care about it – for me reduction in clinical hard end-points is the most important finding, because it proves that science can be translated to clinical benefits and can be translated also into everyday clinical practice. And for me most important finding from our small and surely not perfect study is that using only behavioral intervention we achieved an average 6.0% reduction from baseline body weight and in 61% of participants this reduction exceeded 5.0% (in mentioned above trials with highly expensive tirzepatide there was 76%).

I performed a recommended by you analysis of covariates associated with improvement of metabolic biomarkers (total and LDL cholesterol, triglycerides and uric acid – there was no improvement in plasma glucose and HDL cholesterol levels) and weight loss was the only parameter independently associated with these changes, also after adjustment to age, sex and baseline prevalence of hypertension and dysglycemia The same applies to metabolic indices.

Abstract

  • Please mention that this is a pre-post study design in the abstract.

Response.

Thank you for this remark. We included such a description in the abstract.

Methodology

  • A reference for the definition of overweight and obesity (BMI>25) is needed.

Response.

We included WHO definition of obesity and overweight (the Reference number [21]). By the way – I have read in my life a huge number of papers where obesity was one of the analyzed variables and/or end-point, and in vast majority of them reference for obesity/overweight definition was not present in the Reference list (perhaps no one required that). But I followed this recommendation.

  • Not all the listed medical histories were used for statistical analysis. It might be better to say “used to check for eligibility”??.

Response.

Thank you for this remark. We fully agree with this suggestion and we changed the text accordingly.

  • Equipment used to measure blood pressure, and the cutoffs applied to define hypertension should be mentioned.
  • Additionally, please detail the bioindicators and cutoffs applied to define dyslipidaemia, and dysglycemia (prediabetes or diabetes), including the references used.

Response.

We have added a description of the equipment used to measure blood pressure. Regarding the last part – we were not aimed to diagnose hypertension – the participants in our study were under the care of their family doctors and they were responsible for the diagnosis and treatment of hypertension. We only collected information about which of the study participants is treated for hypertension and which is not. The same applies to dyslipidemia.

  • The authors state that the adherence to the intervention was assessed by interviewing participants, but it is unclear how these interviews were conducted, and what questions were asked? This should be clarified or, at the very least, addressed in the limitation section if there is no standardized protocol, including its potential impacts on the results. Same applies to the physical activity assessment.

Response.

In the Methodology section we added more detailed description how the adherence to diet was assessed. We fully agree that lack of use of standardized protocol is the limitation of our study and we added such a statement in the discussion. However, on the other hand, the best prove of adherence to dietary recommendations was the weight loss achieved by study participants. We also have clarified the description of the physical activity sessions, which clearly states that all training sessions were conducted in a medical center under the supervision of a physiotherapist.

Results

  • It is very uncommon for a large section of the results to be based on findings from supplementary materials. If the authors decide to report these findings, perhaps they could include one table in the main manuscript, e.g., report only the differences and p values for the subgroups.

Response.

Thank you for this comment. We replaced detailed description of findings in each subgroup by a short description of significant differences between them.

Discussion

  • - Lines 333-335 – There appears to be an error in this sentence. I assume the authors mean to say: “…..None of these studies analyzed metabolic biomarkers other than anthropometric indices (e.g., body weight and BMI) and hypertension. Metabolic indicators such as lipid profiles, blood glucose, and uric acid levels were not assessed”. ?
  • For Lines 335-336 – As far as I know, that is not the only study. For instance, the paper published by the same authors for reference [16], they assessed the impacts of the same interventions reported in [reference 16] (DASH diet alone and in combination with exercise and caloric restriction) on insulin sensitivity and lipids, in addition to blood pressure and cardiovascular markers in Blumenthal et al. (2010) Hypertension. 55(5): 1199–1205. doi: https://doi.org/10.1161/HYPERTENSIONAHA.109.149153

Response.

Thank you for this comment. We removed this statement and replaced with more appropriate. In fact, in the ENCORE Study all these parameters were analyzed, but the results were published in two separate papers: in Archives of Internal Medicine and in Hypertension (we added suggested reference into our reference list).

  • Please kindly check and revise the wording, esp. regarding terms such as ‘first study’ or ‘only study’.

Response.

Thank you for this remark. We followed your recommendation and changed wording.

  • I also recommend a more detailed discussion on the limitations of the study design (pre-post), including how this may have impacted the results.

Response.

Thank you for this comment. We followed your recommendation and we included this problem among limitations.

In my final word I want to thank you for the insightful review. I understand that you are focused on statistics and this is your main point of you. But I am mostly focused on clinical relevance of our results. And this is the main difference between us. I hope that these corrections I made are enough, because I have no more time for making further analyzes – I have my clinical, academic, administrative, family and other duties. And if you are going to order me further changes, I am ready to withdraw our manuscript from Nutrients. And I am sure that we will publish it elsewhere, with similar IF, and maybe even without APC.

Sincerely,

The Corresponding Author

Reviewer 2 Report

Comments and Suggestions for Authors

The authors have made the suggested changes to the text. However, they missed at least one error on line 116 where they reported the stature in m while indicating cm as the unit of measurement. I think that the stature be reported in cm and the entire text be rechecked.

Comments on the Quality of English Language

None

Author Response

Reply to Reviewer 2

Reviewer’s 2 comment

The authors have made the suggested changes to the text. However, they missed at least one error on line 116 where they reported the stature in m while indicating cm as the unit of measurement. I think that the stature be reported in cm and the entire text be rechecked.

Response.

Thank you for this remark. Sorry – I simply missed it. I have corrected the mistake. I have re-checked the entire text for linguistic and grammatical correctness and corrected it with the help of my colleague, an English teacher who has lived in the US for several years.

Thank you for the review and for remarks and suggestions which - hopefully - improved quality of our manuscript.

Reviewer 3 Report

Comments and Suggestions for Authors

Revision 8/21/24

Let me say that my tone was not meant to be offensive, so for that, I am sorry. My comments are designed to provide an objective review, and provide helpful suggestions. Again, my apologies.

The authors did a nice job of responding to many of my comments and made revisions accordingly. There is some need for altered or additional wording to clarify some points.

Introduction

1.      I think that the Introduction can be strengthened by clearly stating the scientific gap in the literature and then how this study fills that gap. That is what I meant by stating that I don’t see the need for this study. It appears that the authors are stating that the need for this study is that previous DASH diet studies have not included a PA component that is supervised and that this is the need for the current study. If that is the case, I suggest that this information be clearly added to the Introduction. I will say that I think a stronger case for the need for the study should be provided.

2.      Lines 42 – 44. I encourage the authors to consider softening the language here. Eating and exercise are “two” of the most important contributors.

3.      Line 85. Only “also” or “additional” is needed here at they mean the same thing.

Methods

4.      I believe readers will be interested in the specific recruitment strategies used within the health care setting. For example, how would one express interest in the study – did they have to express this interest directly to the PCP? Did they receive information from a flyer? from the PCP staff? Was there a telephone screening – if so, what was asked? When and where were the assessments taken? When were the assessments conducted during this process? And further details that walk the reader from the recruitment activities to enrollment in the study.

5.      While the diagram provided is helpful, a consort diagram is a fairly standard way of illustrating participant flow.

6.      I suggest moving the information in Lines 115 – 121 to the Results section.

7.      I appreciate that the authors are interested in ensuring that the findings are applicable across subgroups. However, I am concerned that the sample size, and in some cases the sample size difference (e.g. men and women), do not really provide an accurate assessment of subgroup findings.

8.      Did the interventionist document adherence in any standard and reportable way?

9.      Concerning a power analysis, the authors note that they performed power analysis for each calculation and marked those calculations with sufficient power with a symbol. This information, as well as the statistic used to determine the power should be reported in the statistical analysis section. Also, it will be helpful if the authors note that the decision for the sample size was not based on an a priori power analysis, but on an assumption that power would be sufficient for the analyses.

Results

10.  I suggest summarizing all non-significant findings in 1 succinct sentence.

11.  Consider reporting all results to 1 decimal, with the exception of result between 1 and -1.

12.  I think that the results related to subgroup analyses should be limited to the findings that have sufficient power.

Discussion

13.  The Discussion is improved.

14.  I think that the Discussion of the subgroup findings should be limited to the findings that have sufficient power.

Comments on the Quality of English Language

There are a few, minor grammatical errors. 

Author Response

Reply to Reviewer 3

Reviewer 3

Revision 8/21/24

Comment 1. Let me say that my tone was not meant to be offensive, so for that, I am sorry. My comments are designed to provide an objective review, and provide helpful suggestions. Again, my apologies.

Response 1. Thank you very much for what you wrote. I accept your apology, of course. And to be honest, I am a bit ashamed now that I accused you of being offensive. Why? Because you are one of the best reviewers I have met in my career. Your insightful and constructive comments, remarks, suggestions and recommendations have made our work – in my subjective opinion – much better (regardless of whether it is accepted or not). I greatly appreciate your contribution and am deeply grateful for your influence on the quality of our manuscript. One more thing – I do not write these words to buy your sympathy. It is simply true.

Comment 2. The authors did a nice job of responding to many of my comments and made revisions accordingly. There is some need for altered or additional wording to clarify some points.

Response 2. Thank you for this opinion. I have re-checked the entire text for linguistic and grammatical correctness and corrected it with the help of my colleague, an English teacher who has lived in the US for several years. Hopefully, it looks better now.

Introduction

  1. I think that the Introduction can be strengthened by clearly stating the scientific gap in the literature and then how this study fills that gap. That is what I meant by stating that I don’t see the need for this study. It appears that the authors are stating that the need for this study is that previous DASH diet studies have not included a PA component that is supervised and that this is the need for the current study. If that is the case, I suggest that this information be clearly added to the Introduction. I will say that I think a stronger case for the need for the study should be provided.
    • Thank you for this suggestion. I fully agree with you. I followed your recommendation and added an additional paragraph dedicated to this problem..
  2. Lines 42 – 44. I encourage the authors to consider softening the language here. Eating and exercise are “two” of the most important contributors.
    • Thank you for this comment. I did it.
  3. Line 85. Only “also” or “additional” is needed here at they mean the same thing.
    • Thank you for this remark. Done.

Methods

  1. I believe readers will be interested in the specific recruitment strategies used within the health care setting. For example, how would one express interest in the study – did they have to express this interest directly to the PCP? Did they receive information from a flyer? from the PCP staff? Was there a telephone screening – if so, what was asked? When and where were the assessments taken? When were the assessments conducted during this process? And further details that walk the reader from the recruitment activities to enrollment in the study.
    • Thank you for this suggestion. We expanded description of recruitment process.
  2. While the diagram provided is helpful, a consort diagram is a fairly standard way of illustrating participant flow.
    • Thank you for this remark. I changed the layout of diagram according to your suggestion.
  3. I suggest moving the information in Lines 115 – 121 to the Results section.
    • I agree with this suggestion. And we moved it into Results Chapter.
  4. I appreciate that the authors are interested in ensuring that the findings are applicable across subgroups. However, I am concerned that the sample size, and in some cases the sample size difference (e.g. men and women), do not really provide an accurate assessment of subgroup findings.
    • Thank you for this remark. I am aware of this shortage and I fully agree that sample size, especially in case of such huge imbalance between males and females in this study make our calculations not fully reliable. I added such note into this sub-chapter
  5. Did the interventionist document adherence in any standard and reportable way?
    • I do not feel fully competent to answer this question. However, the first author, who participated directly in the interventional part of this work, provided information that all procedures (especially in the field of physical exercises) were performed in accordance with applicable regulations and standards
  6. Concerning a power analysis, the authors note that they performed power analysis for each calculation and marked those calculations with sufficient power with a symbol. This information, as well as the statistic used to determine the power should be reported in the statistical analysis section. Also, it will be helpful if the authors note that the decision for the sample size was not based on an a priori power analysis, but on an assumption that power would be sufficient for the analyses.
    • Thank you for this comment. We added such information. And regarding method – my statistical program calculates statistical power automatically. And because vast majority of calculations had required statistical power, we marked in tables only those with insufficient power.

Results

  1. I suggest summarizing all non-significant findings in 1 succinct sentence.
    • Thank you for this suggestion. We shortened or – in some cases – removed description of non-significant findings.
  2. Consider reporting all results to 1 decimal, with the exception of result between 1 and -1.
    • Thank you for this comment and suggestion. We limited larger numbers to 1 decimal place, but – due to subtle differences in some cases – we left two decimal places in numbers lower than 10.
  3. I think that the results related to subgroup analyses should be limited to the findings that have sufficient power.
    • Thank you for this suggestion. We limited subgroups analysis description to the findings with sufficient statistical power.

Discussion

  1. The Discussion is improved.
  2. I think that the Discussion of the subgroup findings should be limited to the findings that have sufficient power.
    • Thank you for your opinion. We followed your recommendation.

Comments on the Quality of English Language

There are a few, minor grammatical errors. 

Response. I have re-checked the entire text for linguistic and grammatical correctness and corrected it with the help of my colleague, an English teacher who has lived in the US for several years. Hopefully, it looks better now.

In my last word I want to thank you once again for your work and for huge impact on the quality of our manuscript.

Round 3

Reviewer 3 Report

Comments and Suggestions for Authors

The authors provided a revised manuscript that have helped to strengthen the manuscript. 

The number of participants who were available for end of study measurements should be provided in the chart, with an arrow below the number enrolled. 

Comments on the Quality of English Language

Few minor grammatical errors that can be corrected with copyediting. 

Author Response

Dear Reviewer,

Thank you for your short and clear comment and recommendation. We followed them and we changed the figure as you recommended.

Sincerely,

The Corresponding Author